

# Agrochemical control of gene expression using evolved split RNA polymerase. II

Yuan Yuan[1] and Jin Miao[2]

[1] Department of Neurophysiology and Neuropharmacology, Institute of Special Environmental Medicine and Co-innovation Center of Neuroregeneration, Nantong University, Nantong, China
[2] Duke Kunshan University, Kunshan, Jiangsu Province, China

## ABSTRACT

Agrochemical inducible gene expression system provides cost-effective and orthogonal control of energy and information flow in bacterial cells. However, the previous version of Mandipropamid inducible gene expression system (Mandi-T7) became constitutively active at room temperature. We moved the split site of the eRNAP from position LYS179 to position ILE109. This new eRNAP showed proximity dependence at 23 °C, but not at 37 °C. We built Mandi-T7-v2 system based on the new eRNAP and it worked in both *Escherichia coli* and *Agrobacterium tumefaciens*. We also induced GFP expression in *Agrobacterium* cells in a semi-*in vivo* system. The modified eRNAP when combined with the leucine zipper-based dimerization system, behaved as a cold inducible gene expression system. Our new system provides a means to broaden the application of agrochemicals for both research and agricultural application. Portions of this text were previously published as part of a preprint (https://www.biorxiv.org/content/10.1101/2024.04.02.587689v1).

## INTRODUCTION

Agrochemicals are key to health promotion and growth management in modern agriculture. The development of biosensors for agrochemicals opens new avenues to control cell behavior through agrochemicals applied by remotely piloted aircraft (*Park et al., 2015*; *Zimran et al., 2022*; *Park et al., 2024*). One of the possible applications is to use agrochemicals to control gene expression of bacteria associated with plants or animals. Mandipropamid is an oomycete fungicide (*Blum et al., 2010*). We recently reported the development of Mandipropamid inducible gene expression system (Mandi-T7) using a protein proximity detection platform based on evolved split T7 RNA polymerase (eRNAP) (*Yuan & Miao, 2022*). Our Mandi-T7 system is based on two systems developed by other groups. The first system is the repurposed plant Abscisic acid receptor proteins (PYR1^MANDI and ABI), which will sense agrochemical Mandipropamid and will be brought into proximity upon Mandipropamid binding *via* molecular ratchet mechanism (*Park et al., 2015*; *Zimran et al., 2022*; *Steiner et al., 2023*; *Park et al., 2024*). This in turn activates the eRNAP and leads to expression of genes driven by the T7 promoter (Fig. 1A; *Yuan & Miao, 2022*). The second system we borrowed is the eRNAP system. Wild type T7 RNAP split at site 179 is proximity-independent and was used to construct transcriptional logic gates (*Shis &*

Corresponding author
Jin Miao,
jin.miao@dukekunshan.edu.cn

*Bennett, 2013*; *Segall-Shapiro et al., 2014*). The simultaneous assembly of the split T7 RNAP was abolished through directed evolution, and the eRNAP became proximity-dependent (*Pu, Zinkus-Boltz & Dickinson, 2017*). Several small molecule inducible expression systems were developed based on this eRNAP platform, such as Abscisic acid and Rapamycin (*Pu, Kentala & Dickinson, 2018*), Mandipropamid (*Yuan & Miao, 2022*), and Rapalog (*Martin et al., 2023*). The molecular variety of this eRNAP biosensor platform has been further expanded by fusion with the variable domains of antibodies (*Komatsu, Ohno & Saito, 2023*) or with cell-pole organizing proteins to achieve asymmetric gene expression (*Lin et al., 2021*).

The Mandi-T7 system was initially tested at 37 °C. The failure of Mandi-T7 to work at room temperature hampers engineering initiatives in other bacteria. Here, we report the alleviation of this issue by adopting a new split site of T7 eRNAP. The new system Mandi-T7-v2 works at 23 °C in *E. coli in vitro* and in *Agrobacterium in vitro* and semi-*in vivo*. The modified eRNAP is also compatible with a leucine zipper peptide-based dimerization system and may be used with other chemical-induced dimerization systems as well.

## MATERIAL AND METHODS

### Plasmid construction

The eRNAP$_N$ (derived from 1-109 aa of T7 RNAP), the T7 RNAP$_C$ (110-883 aa of T7 RNAP), and ZA/ZB fragments were synthesized and cloned into the plasmid pJM1B6 (*Yuan & Miao, 2022*) by GenScript Biotechnology (Nanjing, China). The pVS1 origin from pCAMBIA1301 (GenBank: AF234297.1, 2488 bp -6266 bp), the Mandi-T7-v2 driver cassette, and the T7p::mcherry / toehold switch -sfGFP effector cassette were assembled into the pGM1190 plasmid (Addgene #69994) backbone to enable single-plasmid expression in *Agrobacterium*. Detailed information on these genetic parts and plasmids is listed in Tables S1 and S2.

### Mandipropamid responsive assay for evolved split T7 RNA polymerase

The response to Mandipropamid was assayed, as previously reported. Briefly, the strain Top10 (TIANGEN Biotech, Beijing, China) was transformed with the driver and reporter plasmids. Single colonies were inoculated into SOC medium supplemented with Ampicillin (100 mg/mL) and Spectinomycin (50 mg/mL), and the mixture was allowed to grow at 37 °C. The overnight culture was transferred to a fresh medium with antibiotics at a 1:400 ratio and incubated for 3 h at 37 °C. Then, induction by Mandipropamid was tested at both 23 °C and 37 °C. Mandipropamid (sc-235565; Santa Cruz) was added as the inducer, and DMSO as the solvent control. After incubating for 3 h at 37 °C or 6 h at 23 °C, 100 μL of each sample was transferred to a 96-well plate. The fluorescence signal (GFP, Ex: 488 nm, Em: 510 nm; mcherry, Ex: 587 nm, Em: 630 nm) and OD$_{600}$ of the culture were then measured using a Thermo Fisher Varioskan LUX plate Reader. For *Agrobacterium tumefaciens*, the strain LBA4404 (Weidibio, Shanghai, China) was transformed with the plasmid pJM-Mandi-T7-Ag and grown at 28 °C. Single colonies were inoculated with SOC medium supplemented with Apramycin (50 mg/mL) and grown at 28 °C. The overnight culture was transferred to a fresh medium with antibiotics at a 1:100

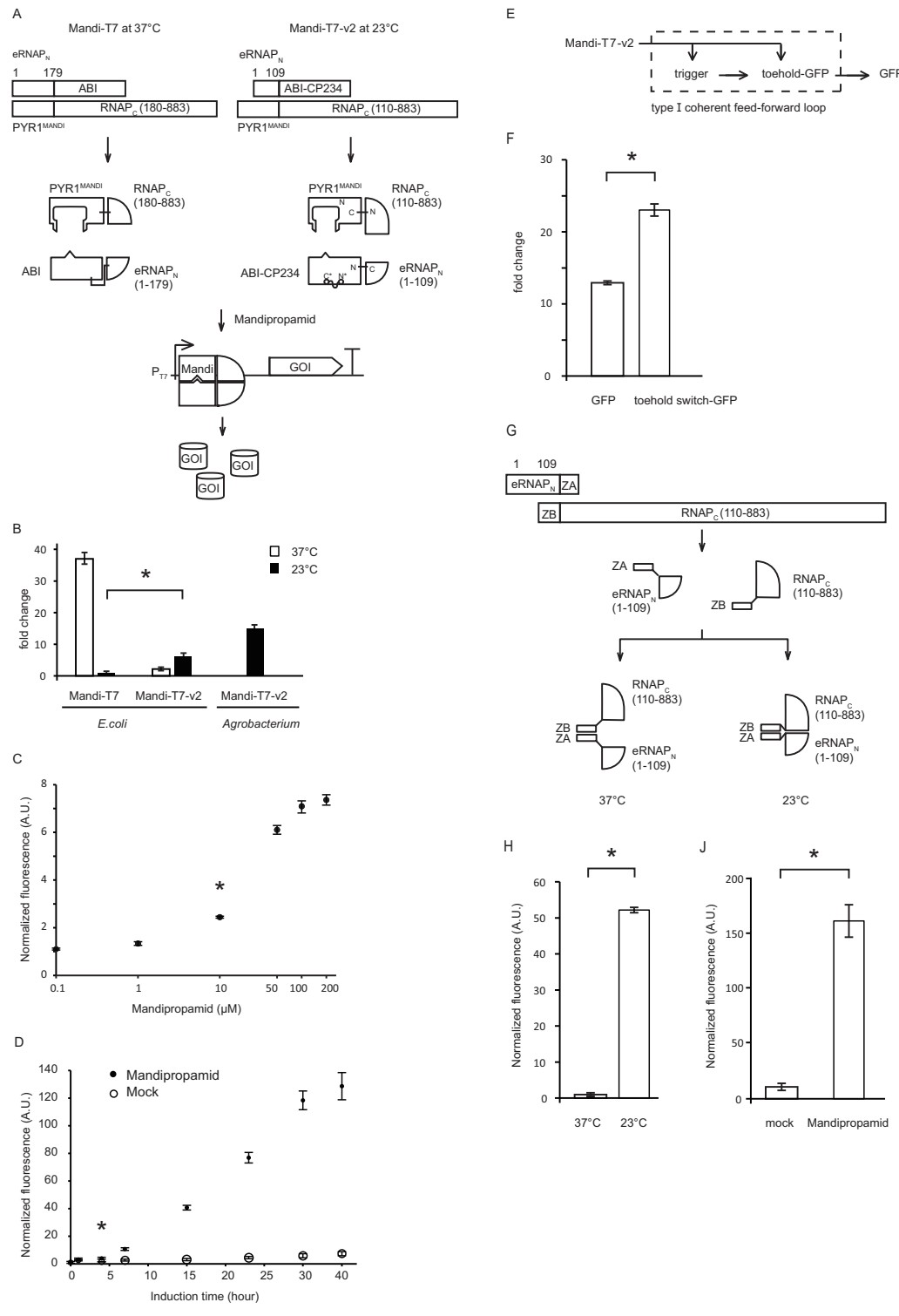

**Figure 1** (A–H) Engineering the Mandi-T7-v2 system.

ratio and incubated for 3 h at 28 °C. Induction was performed overnight at 23 °C after adding Mandipropamid or DMSO (vehicle control). For both *E. coli* and *Agrobacterium tumefaciens*, the fluorescence intensity was divided by or normalized with the $OD_{600}$ value. Then, normalized fluorescence intensity values from induced samples (Mandipropamid) and control (DMSO) were compared to yield fold change. Raw data was provided as a Supplemental File. The reporter gene mcherry was used for Figs. 1B, 1C, and 1D. The reporter gene GFP was used for Figs. 1F, 1H, and 1J.

## Cold responsive assay of split eRNAP fused with leucine zipper peptides

The *E. coli* strain Top10 was used. Single colonies were inoculated with SOC medium supplemented with Ampicillin (100mg/mL) and Spectinomycin (50 mg/mL) and grown at 37 °C. The overnight culture was transferred to a fresh medium with antibiotics at a 1:400 ratio and incubated for 3 h at 37 °C. Following this, half the culture was incubated at 23 °C, the other half stayed at 37 °C. After 3 h at 37 °C or overnight at 23 °C, 100 µL of each sample was transferred to a 96-well plate. The florescence signal (GFP, Ex: 488 nm, Em: 510 nm) and $OD_{600}$ of the culture were measured using a Thermo Fisher Varioskan LUX plate Reader. Raw data was provided as a Supplemental File.

## Time course fluorescence measurement

*E. coli* Top10 strain containing Mandi-T7_v2 and reporter plasmids was generated as mentioned above. Single colonies were used to inoculate SOC medium supplemented with Ampicillin (100 mg/mL) and Spectinomycin (50 mg/mL) and grown at 37 °C overnight. The Overnight culture was transferred to a fresh medium with antibiotics at a 1:100 ratio. Afterwards, either Mandipropamid (50 mM, sc-235565; Santa Cruz) or DMSO (solvent control) was added at a 1:1000 ratio. The culture was incubated at 23 °C for 40 h. Samples were harvested after 1 h, 4 h, 7 h, 15 h, 23 h, 30 h and 40 h. The fluorescence (mcherry: Ex: 587 nm, Em: 630 nm) and $OD_{600}$ of 100 µL of each sample were measured using a Thermo Fisher Varioskan LUX plate Reader. Raw data was provided as a Supplemental File.

## Mandipropamid response assay in a semi-*in vivo* system

*Agrobacterium tumefaciens* strain LBA4404 containing toehold switch-GFP driven by Mandi-T7-v2 was generated as mentioned above. Single colonies were inoculated LB medium with antibiotics at 28 °C. Overnight culture was transferred to a fresh medium with antibiotics at a 1:100 ratio. When $OD_{600}$ reached roughly 0.4, aliquots of 300 µL culture were added into the hollow septate stems of water spinach (*Ipomoea aquatica*), which were cut into cylinder shaped and inserted into a 96-well deep well plate as previously reported (*Yuan & Miao, 2022*). Mandipropamid (200 µM final) or DMSO (solvent control) was added. The plate was shaken in an orbital shaker (1,000 rpm) at 23 °C for 24 h. A total of 200 µL of each sample was added to a 96-well plate. Both fluorescence (GFP: Ex: 488 nm, Em: 510 nm;) and $OD_{600}$ were measured by a Thermo Fisher Varioskan LUX plate Reader. The fluorescence intensity was normalized with the $OD_{600}$ value. Values from induced samples (Mandipropamid) and control (DMSO) were compared to yield fold change. Raw data was provided as a Supplemental File.

### Statistical analysis

Datasets were analyzed within the Office Excel software. Two-tailed t-tests were used for pairwise comparisons. *P* values were indicated in the figure legend.

## RESULTS

To apply Mandi-T7 to bacteria living at room temperature, we characterized Mandi-T7 at 23 °C. The results showed that the fluorescence signal of the report gene could be detected regardless of the presence of the inducer (Fig. 1B and Fig. S1), indicating that the eRNAP relapsed into self-assembly at 23 °C. Restoration eRNAP's proximity dependency at 23 °C will enable Mandi-T7 to work at 23 ° C. We speculated that a new split site in the N terminal region of T7 RNA polymerase, not previously reported by the bisection mapping efforts (*Segall-Shapiro et al., 2014*), might prevent self-assembly at 23 °C. We scrutinized the structure of the T7 RNAP N terminal region. We selected ILE109 as our candidate because ILE109 is exposed at the surface and splitting at ILE109/LYS110 will break the connection between the Arginine loop, which is essential for binding upstream AT-rich region of the T7 promoter, and the rest of the T7 RNA polymerase (Fig. S2).

To assess the potential improvement in induction at 23 °C, we modified the driver module of the Mandi-T7 system (Fig. 1A). The split site was changed to ILE109/LYS110 while retaining the point mutations in the N-terminal region of the eRNAP (F21L, L32S, E35G, R57C, E63K, K98R, Q104K, Q107K) as leverage for flexibility. Induction for 6 h at 23 °C yielded a modest fold change over the control (Fig. 1B). Unexpectedly, induction by Mandipropamid at 37 °C was abolished by adopting the new split-site (Fig. 1B). Encouraged by these findings, we named this modified eRNAP as eRNAP2, and our new Mandipropamid inducible expression system Mandi-T7-v2. We proceeded to characterize the induction with different concentrations of Mandipropamid. We observed maximum induction by 200 µM Mandipropamid at 23 °C (Fig. 1C). We further evaluated the kinetic characteristics of the Mandi-T7-v2 at 23 °C over 30 h. We observed an induction signal after 1 h and a continuous increase over 30 h (Fig. 1D and Fig. S1). However, the fluorescence signal of the non-induction samples accumulated simultaneously, resulting in marginal increase in fold induction after 23 h (Fig. 1D).

To increase the dynamic range without further engineering the Mandi-T7-v2 system, we tried to incorporate the toehold switch into the reporter module (Fig. 1E), which has been shown to improve the performance of T7-based inducible expression systems through a coherent type 1 feed-forward loop (*Hwang et al., 2021*; *Greco et al., 2021*). Indeed, after 24 h induction, incorporation of the toehold switch could increase the dynamic range by two-fold (Fig. 1F).

To test whether Mandi-T7-v2 can be applied to other bacteria, we tried Mandi-T7-v2 in *Agrobacterium tumefaciens*, a plant pathogen and medium of T-DNA transformation. To generate a single-plasmid-based induction system, we assembled the Mandi-T7-v2 driver module, the reporter module and the pVS1 replication origin together. We tested the Mandipropamid induction in *Agrobacterium* at 23 °C. The result showed a 15-fold induction (Fig. 1B and Fig. S1), suggesting that the Mandi-T7-v2 system also works in other bacteria.

To test whether induction at 23 °C but not at 37 °C is an inherent property of the new eRNAP2 or depends on the fusion to the ABI-PYR1$^{MANDI}$ protein pair, we replaced the ABI-PYR1$^{MANDI}$ pair with leucine zipper peptides ZA and ZB (Fig. 1G), one of the model dimerization systems that will lead to spontaneous dimerization and has been used for developing the evolved RNAP system before (*Pu, Zinkus-Boltz & Dickinson, 2017*; *Pu, Kentala & Dickinson, 2018*). We tested the ZA/ZB-eRNAP2 system at both 23 °C and 37 °C. The results showed 50-fold higher expression at 23 °C over 37 °C (Fig. 1H and Fig. S1), reminiscent of a cold inducible expression system. This result indicates the restored proximity dependency of the eRNAP2 at 23 °C does not require specific fusion partners.

Encouraged by the results above, we evaluated the ability of Mandi-T7-v2 to function in plants at 23 °C. We used the hollow septate stems of water spinach (*Ipomoea aquatica*) as the containers to induce bacterial gene expression as previously reported (*Yuan & Miao, 2022*). We used toehold switch-regulated GFP as the report gene. *Agrobacterium* culture containing Mandi-T7-v2 and the reporter gene was incubated in the water spinach stems at 23 °C for 24 h. The results showed 15-fold induction (Fig. 1J) over vehicle control. This result suggests Mandi-T7-v2 might be useful in other complex settings such as vector-borne bacterial plant pathogens.

## DISCUSSION

Fortunately, we have found a new split site of eRNAP, which enables chemical inducible activity control at 23 °C. Induction around 23 °C will be an advantage for controlling gene expression in bacteria living around room temperature, especially bacteria associated with model organisms raised around 23 °C, such as *Arabidopsis*, *Drosophila*, and *C. elegans*. As a proof-of-principle, we have shown that reported gene expression could be induced in *Agrobacterium* inside plant tissue. Our new system also widens the temperature range of the existing induction system. For example, proteins produced by Mandi-T7 system in *E. coli* may not be properly folded at 37 °C. Misfolding may be mitigated at 23 °C in our new system. This new eRNAP2 system can also be adapted to other proximity-dependent systems. Our cold response assay of the ZA/ZB-eRNAP2 system exemplified this possibility. For dimerization pairs that work at 23 °C but not at 37 °C, eRNAP2 would be preferred as their interaction sensor.

When we tested the performance of the Mandi-T7 system at room temperature, we found constitutive activity without inducibility, indicating the disrupted spontaneous assembly of eRNAP at 37 °C was restored at room temperature. The temperature sensitivity of the eRNAP platform may be explained by its origin in the directed evolution system running at 37 °C (*Pu, Zinkus-Boltz & Dickinson, 2017*).

Several split sites of T7 RNA polymerase have been identified by the bisection mapping, like around 67, 301, 563, and 763 (*Segall-Shapiro et al., 2014*). Except for split site 563, which has also been used for engineering the thermo-repressible split T7 RNAP (*Chee et al., 2022*), and the blue light-inducible systems (*Han, Chen & Liu, 2017* and *Dionisi et al., 2022*), these split sites have not been developed into general or specific biosensor platforms. Our work suggests it is worth exploring new split sites of T7 RNA polymerase for specific applications.

We incorporated a toehold switch into our simple gene circuit to improve dynamic range. This provided an easy way to improve the performance of Mandi-T7-v2 system on protein-encoding genes. However, the leakage issue persists for RNA devices driven by Mandi-T7-v2. Further directed-evolution efforts may be required to reduce the background activity of the Mandi-T7-v2 system.

The ABI-PYR1 pair has been recently expanded to be sensors of other agrochemicals (*Zimran et al., 2022*; *Park et al., 2024*). Our work will readily help develop new inducible gene expression systems of these agrochemicals. Both the plant growth-promoting (PGP) traits and synthetic bacteria-to-plant communication pathways have been engineered in the root-associated bacteria (*Haskett, Tkacz & Poole, 2021*; *Boo et al., 2024*). Key PGP traits include nitrogen fixation, nutrient mobilization, antibiotics production, *etc.* (*Haskett, Tkacz & Poole, 2021*). Agrochemical control of PGP traits or bacteria-to-plant communication will add temporal modulation to these devices.

## ACKNOWLEDGEMENTS

We would like to thank Professor Linfeng Huang (Duke Kunshan University) for his discussions and generous sharing equipment and reagents. We also want to express our gratitude to Dr. Jinyue Pu and Professor Bryan Dickinson of the University of Chicago for their sharing plasmid information of the original eRNAP system.

### Funding

The authors received no funding for this work.

### Competing Interests

The authors have filed a patent application for the technology described in this article (application number: 2024102651123).

### Author Contributions

- Yuan Yuan performed the experiments, analyzed the data, authored or reviewed drafts of the article, and approved the final draft.
- Jin Miao conceived and designed the experiments, performed the experiments, analyzed the data, prepared figures and/or tables, authored or reviewed drafts of the article, and approved the final draft.

### Data Availability

The sequence of the genetic parts and the raw data are available in the Supplemental Files.

### Supplemental Information

Supplemental information for this article can be found online at http://dx.doi.org/10.7717/peerj.18042#supplemental-information.

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
