# Peer review of "Agrochemical control of gene expression using evolved split RNA polymerase. II"

_PeerJ, doi:10.7717/peerj.18042_

## Round 0.1 · original submission · Major Revisions

Please attend to the reviewer's comments in a detailed point-by-point rebuttal letter.

Reviewer 1 ·

Basic reporting

Here, the authors report a modified system for agrochemical control of gene expression using evolved split RNA polymerase II. In general, much information is missing in all sections of the manuscript. The objective of the work is not entirely clear and the experiments are very preliminary.

Minor comments:
Line 20: Please correct Agrobacterium
Line 28: “Mandipropamid is an oomycete fungicide” reference is missing.
Line 29-31: You didn’t cite your work.
Line 31: Why ABA?
Line 36: You already defined eRNAP
It is correct to write RNA polymerase. II?

Major comments:
In the introduction section, it isn't very clear which data was generated by you and which by other authors. It would help if you were more clear in your writing.

Please explain which genes could control this system.

Please explain how this system could control via remote the cell behavior.

Please explain how controlling the gene expression of bacteria associated with plants can help plant health. Which are the mechanisms?

Your discussion section is very poor, you need to explain in more details your results.

Experimental design

Many important points are not described in the Material and Methods section.

It is not reported how the fold change was calculated. Which was the reported gene? GFP? You didn't describe any of this experiment.

In Figure 1F, what are the differences versus 1B in addition to the temperatures?

Why did you perform the "cold responsive assay" at 23 and 37ºC?

Validity of the findings

No statistical analyses were performed to support the data

Additional comments

I consider that the results shown here are not sufficient. The developed system should be tested in in vivo conditions in plants to reach an accurate conclusion.

·

Basic reporting

Clear and unambiguous, professional English used throughout:
The manuscript is written in clear and professional English. However, minor grammatical and syntactical errors should be addressed to enhance readability. For example, “Agrobacteria tumefaciens” should be “Agrobacterium tumefaciens” for consistency and accuracy.

Literature references, sufficient field background/context provided:
The article provides sufficient background and context, referencing relevant prior literature. The introduction clearly situates the work within the broader gene expression systems and agrochemical applications field. However, it would benefit from a more detailed discussion of the limitations of the previous systems and a stronger rationale for the new approach.

Professional article structure, figures, tables. Raw data shared:
The manuscript follows a standard scientific structure with clearly defined sections. Figures are relevant, appropriately described, and labelled, though some could benefit from higher resolution or clearer legends. The raw data appears to be shared following data-sharing policies, but explicit confirmation or access details should be provided.

Self-contained with relevant results to hypotheses:
The manuscript is self-contained and presents a coherent work addressing the stated hypotheses. The results are relevant to the hypotheses and are presented in a logical sequence.

Experimental design

Original primary research within Aims and Scope of the journal:
The research is original and fits within the aims and scope of the journal. The research question is well-defined, relevant, and meaningful. The study addresses a significant knowledge gap by improving the temperature sensitivity of the Mandi-T7 system.

Rigorous investigation performed to a high technical & ethical standard:
The investigation appears to be conducted rigorously and to a high technical standard. However, the manuscript would benefit from more detailed descriptions of the controls used in the experiments, particularly in the Mandipropamid responsive assays.

Methods described with sufficient detail & information to replicate:
The methods are generally described with enough detail to allow replication. However, some procedural steps, such as the exact conditions for directed evolution or specifics of the fluorescence measurements, should be elaborated for clarity.

Validity of the findings

Impact and novelty not assessed. Meaningful replication encouraged where rationale & benefit to literature is clearly stated:
The manuscript does not rely on subjective assessments of impact or novelty. The rationale for replication of previous findings is clearly stated, and the new findings add significant value to the existing literature. The novel split site and its implications for temperature-dependent gene expression are well-justified.

All underlying data have been provided; they are robust, statistically sound, & controlled:
The underlying data appear to be robust, statistically sound, and well-controlled. However, the manuscript should include more detailed statistical analyses of the experimental results to strengthen the validity of the conclusions.

Conclusions are well stated, linked to the original research question & limited to supporting results:
The conclusions are well-stated and linked to the original research question. They are appropriately limited to the supporting results, avoiding overgeneralization. The discussion effectively ties the findings back to the hypothesis and suggests future directions for research.

---

## Round 0.2 · Minor Revisions

Please review the attached PDF for editorial suggestions and submit a revised version at your earliest convenience.

Reviewer 1 ·

Basic reporting

The authors have addressed each of the points indicated. The manuscript was considerably improved and seems optimal for publication.

My only concern is about the figures, they seem low quality; I suggest the authors check the pixels.

Experimental design

All points were attended, and the materials and methods are much clearer.

Validity of the findings

No comment

Additional comments

No comment

---

## Round 0.3 · accepted · Accept

Thank you for addressing the suggested recommendations and style modifications.